# PERSONALVIDEO: HIGH ID-FIDELITY VIDEO CUSTOMIZATION WITH STATIC IMAGES

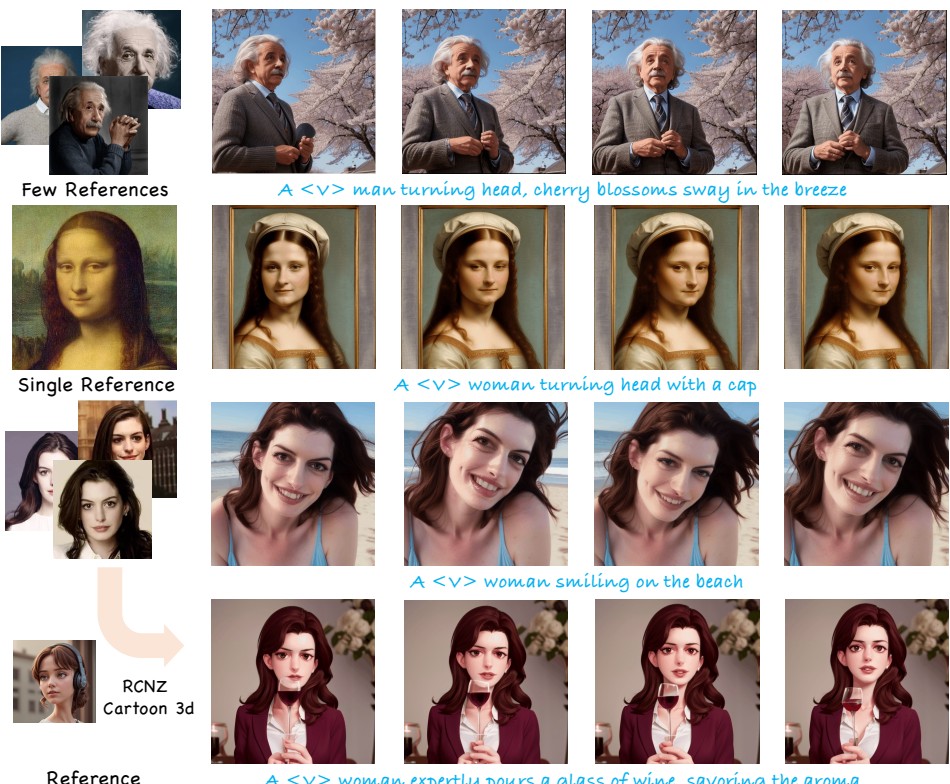

Figure 1: **Results of the proposed PersonalVideo.** Given a few or just one reference image of a specific identity, PersonalVideo can generate temporal consistent videos aligned with the prompt and seamlessly integrate pre-trained component in the AIGC community. Generated video samples are available at `https://personalvideo.github.io/`

## ABSTRACT

The current text-to-video (T2V) generation has made significant progress in synthesizing realistic general videos, but it is still unexplored in identity-specific human video generation with customized ID images. The key challenge lies in maintaining high ID fidelity consistently while preserving the original motion dynamic and prompt following after the identity injection. Current video identity customization methods mainly rely on reconstructing given identity images on text-to-image models, which have a divergent distribution with the T2V model. This process introduces a tuning-inference gap, leading to identity inaccuracy and dynamic degradation. To tackle this problem, we propose a novel framework, dubbed **PersonalVideo**, that applies direct supervision on videos synthesized by the T2V model to bridge the gap. Specifically, we introduce a learnable Spatial Identity Adapter under the supervision of pixel-space ID loss, which customizes the specific identity and preserves the original T2V model's abilities (e.g., motion dynamic and prompt following). Furthermore, we employ simulated prompt aug-

mentation to reduce overfitting by supervising generated results in more semantic scenarios, gaining good robustness even with only a single reference image available. Extensive experiments demonstrate our method's superiority in delivering high identity faithfulness while preserving the inherent video generation qualities of the original T2V model, outshining prior approaches. Notably, our PersonalVideo seamlessly integrates with pre-trained SD components, such as ControlNet and style LoRA, requiring no extra tuning overhead.

# 1 INTRODUCTION

Recently, there has been considerable scholarly interest in text-to-video (T2V) generation (Guo et al., 2023; Wang et al., 2023a; Chen et al., 2024; Brooks et al., 2024), which allows for the production of videos from user-defined textual descriptions. However, the identity-specific customization of high-fidelity human videos has not been fully explored. It aims to customize a wide variety of engaging videos using a few users' photos, allowing for personalized content creation that features them in different actions, scenes, or styles while maintaining high ID fidelity. Without the need for complex scene construction and tedious post-production special effects, this convenient way of video creation also has great potential in the film and television industry.

Identity customization has achieved significant advancements in the field of text-to-image (T2I) (Gal et al., 2022; Ruiz et al., 2023; Ye et al., 2023; Li et al., 2024; Wang et al., 2024b; Guo et al., 2024). Typically, these methods use a reconstructive approach on provided reference images to inject the identity into the pre-trained T2I model during the customization. However, directly employing this strategy in video customization will lead to unsatisfied results due to two notable challenges:

1) **Inserting consistent identity with high fidelity.** Existing video customization methods (Ma et al., 2024; Wei et al., 2024) naively use image reconstruction supervision during tuning to model a customized T2I prior, which is then injected into the T2V model to generate identity-specific videos during inference. However, the distribution of the pre-trained T2V model often deviates from that of the pre-trained T2I model. Fig. 2 shows that the tuning-inference gap will lead to a degradation of learned identity. As humans are sensitive to facial features, higher fidelity and consistent identity are required in generated videos.

2) **Preserving inherent motion dynamics and prompt following.** During the customization, tuning on limited static images will significantly shift the video prior of the pre-trained T2V model, making the generated videos tend to appear static and fail to follow the given prompts. Although some works (Wei et al., 2024; He et al., 2023; Chefer et al., 2024) solve this problem by requiring additional video input, it brings great inconvenience to users. As images from user input contain no video prior, it is important to preserve the original T2V model's abilities.

To address these challenges, we propose a novel framework, dubbed **PersonalVideo**, for ID-specific video generation that can achieve high ID fidelity and maintain original motion dynamics and prompt following with only a few images of an identity given. Based on a pre-trained video generation model, our PersonalVideo injects the identity from given images into some learnable modules through an optimization process. Different from previous supervising this tuning process via reconstructing images on T2I models, our core insight is applying identity supervision directly to videos generated by the T2V model thus bridging the tuning-inference gap as shown in Fig. 2.

Without ground truth for generated videos, we can only use nonconstructive supervision. Inspired by face generation (Wang et al., 2021; Richardson et al., 2021) in the realm of Generative Adversarial Networks (Goodfellow et al., 2020), directly applying human perception loss to the generated videos could reward the T2V model generating videos with high ID fidelity. During the tuning time, we first generate videos from pure noises and then align face embeddings extracted from generated videos and reference images via the identity encoder (Deng et al., 2019). Building on recent advancements in fast sampling methods (Wang et al., 2024a; 2023b) for T2V models, this process only requires four denoising timesteps with manageable tuning costs and negligible quality loss. Furthermore, during this process, we can incorporate simulated prompts augmentation, which supervises generated results in more semantic scenarios. Unlike the reconstructive training method, they are not limited by the number of references, effectively mitigating overfitting and demonstrating strong robustness, even when only a single reference image is available.

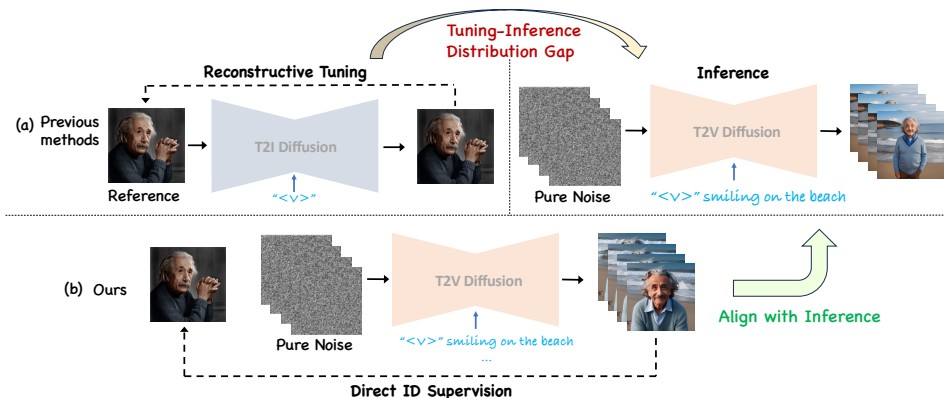

Figure 2: Previous T2V customization methods supervise the tuning process via reconstructing images on T2I models, which suffers from **a tuning-inference gap**. Differently, we aim to directly apply identity supervision on generated videos, which aligns with inference and bridges the gap.

To achieve the identity injection with preserved motion dynamics and prompt following, it is essential to confine the learnable modules to select regions. Through the experiments on T2V diffusion models, we can observe that: (1) In the early stages of the denoising process, the focus is on restoring layout and motion, while the later stages concentrate on recovering detailed object appearance (Cao et al., 2023; Patashnik et al., 2023). (2) The spatial self-attention layers in the pre-trained diffusion model play a crucial role in preserving the geometric and shape details for the identity (Liu et al., 2024), while the spatial cross-attention layers are primarily responsible for preserving semantic information. Based on the observations, we propose the Spatial Identity Adapter, which is injected into the spatial self-attention layer in the only last denoising step of the fast sampling diffusion model.

Qualitative and quantitative experiments demonstrate that our **PersonalVideo** achieves high ID fidelity and effectively preserves the original T2V model's capabilities. Benefiting from the scalability of T2V models, it also supports any style-specific fine-tuned model and other conditional inputs, such as poses, which offer valuable flexibility for abundant creation in the AIGC community. Our contributions are summarized as follows:

- We introduce a novel framework, dubbed **PersonalVideo**, for video personalization with limited images, achieving high ID-fidelity and preserving original motion dynamics and prompt following.
- To bridge the tuning-inference gap, we propose to directly apply identity supervision on the generated videos and employ simulated prompts augmentation to robustly achieve high ID-fidelity, even for just single image.
- We introduce a Spatial Identity Adapter to inject the identity and effectively mitigate the degradation of motion and semantics. It can also seamlessly combine with other pre-trained SD components, without the need for extra tuning effort.

## 2 RELATED WORK

**Text-to-Video Generation.** The topic of T2V generation has attracted considerable interest among researchers for a long time. Recently, the utilization of diffusion models has become predominant in the realm of T2V tasks. VDM (Ho et al., 2022b) stands as the pioneer that first leverages a diffusion model for T2V generation. Subsequently, Make-A-Video (Singer et al., 2023) and Imagen Video (Ho et al., 2022a) were proposed to generate high-resolution videos in pixel space. To save computational resources, various frameworks have been developed to perform a latent denoising process (Zhou et al., 2022; He et al., 2022; Wang et al., 2023a; Blattmann et al., 2023; Wang et al., 2023c; Chen et al., 2023). Although these methods can produce high-quality generic videos by pre-training on large-scale text-video pair datasets, it remains challenging to enable them to synthesize contents according to specified identities.

**Text-to-Image Identity Customization.** In the field of T2I, a lot of approaches have emerged for ID customization. As a seminal work, Textual Inversion (Gal et al., 2022) represents the user-

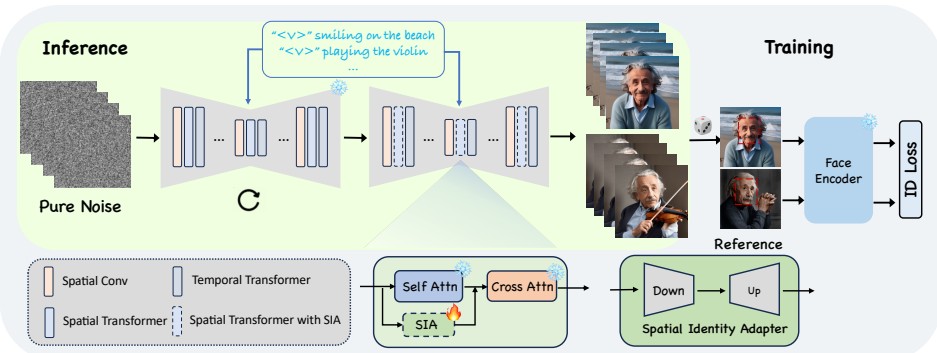

Figure 3: **Overview of PersonalVideo framework.** To bridge the tuning-inference gap, we directly apply ID supervision on generated videos starting from pure noises. During the optimization, we adopt simulated prompt augmentation to supervise generated results in more semantic scenarios. To preserve original motion dynamics and prompt following, we introduce a Spatial Identity Adapter (SIA) to inject the identity into the spatial self-attention layer only for the last denoising steps.

provided identity with a specific token embedding of a frozen T2I model. For better ID fidelity, DreamBooth (Ruiz et al., 2023) further optimizes the original model, where efficient fine-tuning techniques such as LoRA (Hu et al., 2021) can also be applied. On the other hand, several tuning-free methods have been explored, aiming to directly inject ID information into the generation process. IP-Adapter (Ye et al., 2023) and InstantID (Wang et al., 2024b) focus on adapting encoders that extract ID-relevant information. PhotoMaker (Li et al., 2024) proposes to enhance the ID embedding based on large-scale datasets comprising diverse images of each ID. PuLID (Guo et al., 2024) suggests optimizing an ID loss between the generated and reference images in a more accurate setting.

**Text-to-Video Identity Customization.** The T2V customization task presents further challenges compared to the T2I task due to the temporal motion dynamics involved in videos. Currently, only a limited number of works have undertaken early investigations into this area. MagicMe (Ma et al., 2024) adopts an ID module based on extended Textual Inversion. However, the model is trained under T2I reconstruction supervision, which deviates from the T2V setting of inference, leading to inferior ID fidelity. DreamVideo (Wei et al., 2024) can customize the identity given a few images, but the inconvenience lies in the fact that it necessitates additional videos to provide motion patterns. ID-Animator (He et al., 2024) proposes to encode ID-relevant information with a face adapter, which requires thousands of high-quality human videos for fine-tuning, thereby incurring significant costs associated with dataset construction and model training.

## 3 PRELIMINARY

**Text-to-Video Diffusion Models.** Text-to-video diffusion models (T2V) (Blattmann et al., 2023; Guo et al., 2023; Wang et al., 2023a; 2024a) are tailored for generating videos by adapting image diffusion models to handle video data. Specifically, the diffusion model $\epsilon_\theta$ aims to predict the added noise $\epsilon$ at each timestep $t$ based on text condition $c$, where $t \in \mathcal{U}(0, 1)$ is normalized. The training objective can be simplified as a noise-prediction loss:

$$\mathcal{L}_{\text{diff}} = \mathbb{E}_{z, c, \epsilon \sim \mathcal{N}(0, \text{I}), t} \left[ \| \epsilon - \epsilon_\theta \left( z_t, \tau_\theta(c), t \right) \|_2^2 \right], \tag{1}$$

where $z \in \mathbb{R}^{B \times F \times H \times W \times C}$ is the latent code of video data with $B, F, H, W, C$ being batch size, frame, height, width, and channel, respectively. $\tau_\theta$ presents a pre-trained text encoder. A noise-corrupted latent code $z_t$ from the ground-truth $z_0$ is formulated as $z_t = \alpha_t z_0 + \sigma_t \epsilon$, where $\alpha_t$ and $\sigma_t$ are hyperparameters to control the diffusion process.

**ID Customization.** ID customization for text-to-image models (T2I) focuses on enabling pre-trained models to generate images that reflect specific identities while adhering to the provided text descriptions. Typically, they optimize a new word embedding for the user-provided ID or fine-tune the generator to enhance ID fidelity. Similarly to the training objective of pre-trained diffusion models, they adopt the reconstruction supervision with L2 noise-prediction loss to bind the user-provided references with the special token to inject to identity to the pre-trained T2I models.

## 4 METHODOLOGY

Despite the great progress of ID-specific image generation, it is still challenging for T2V customization due to the additional consistency requirements. Given few images for a specific identity, our goal is to generate customized videos with high ID-fidelity and preserve original motion dynamics and prompt following. In this paper, we propose a novel framework, dubbed **PersonalVideo**, for high ID fidelity video customization as shown in Fig. 3. Specifically, we propose a non-reconstructive approach to directly learn the identity as depicted in Sec. 4.1. To inject the identity with preserved motion dynamics and prompt following, we design a spatial identity adapter in Sec. 4.2. Besides, we introduce a simulated prompt augmentation to mitigate overfitting in Sec. 4.3.

### 4.1 NON-RECONSTRUCTIVE T2V CUSTOMIZATION

Current T2V customization methods typically adopt a reconstruction approach to train a customized T2I prior on provided images and inject it into the pre-trained T2V models to generate identity-specific videos. However, it leads to a tuning-inference gap due to the misaligned distribution between reference images in the tuning time and generated videos in inference time, which brings inferior ID fidelity and degradation of inherent motion dynamics and prompt following.

To bridge the gap, as shown in Fig. 2, we propose a non-reconstructive framework to directly apply identity supervision on the generated videos for high ID fidelity with the references. Inspired by face generation (Wang et al., 2021; Richardson et al., 2021) in the realm of GAN (Goodfellow et al., 2020), directly applying human perception supervision to the generated videos could reward the T2V model generating videos with high ID fidelity. Therefore, during the tuning time, we start from pure noise instead of noised references in the previous reconstructive methods. Then we directly supervise the sampled videos to mimic the identity in the generated videos with that in references using ID loss, which closely aligns with human perception and the distribution of the real world.

Specifically, we use pre-trained ID encoder (Deng et al., 2019) $\phi$ to precisely extract the ID embeddings of the references and the random $i$-th frame of the sample video. Then we minimize the cosine similarity of them to align the identity effectively and directly. Here we crop the faces from the images and adopt image augmentation techniques like color jitter to make it more robust for limited references. Formally, we calculate:

$$\mathcal{L}_{\text{id}} = \mathbb{E}_{c,i}\left[CosSim\left(\phi(I_{id}), \phi(\text{F-T2V}(z_T, c, i))\right)\right],  \tag{2}$$

where $I_{id}$ are the reference images and $c$ are the text prompts with the specific keyword.

### 4.2 SPATIAL IDENTITY ADAPTER

To achieve the identity injection with preserved motion dynamics and prompt following, we introduce a Spatial Identity Adapter as shown in Fig. 3, which addresses the degradation of original motion generation and prompt following.

For motion preservation, we conduct exploratory experiments on video diffusion models to investigate the varying focus on motion at different denoising steps. As shown in Fig. 4, we can observe that the motion of the person is formed in the early denoising step. During these steps, the model tends to restore the layout (Cao et al., 2023) and motion. In contrast, the later steps focus on recovering of the detailed appearance of the objects (Patashnik et al., 2023). Based on the observation, we propose to inject the identity only in the later denoising steps during training and inference time as shown in Fig. 3, to reduce the influence on

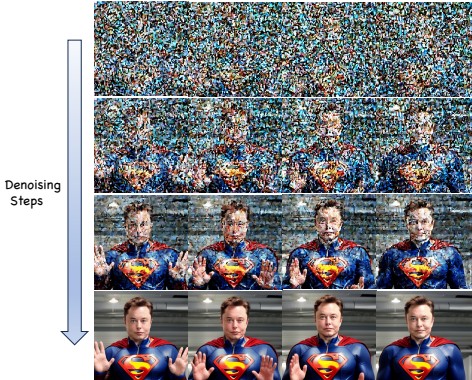

Figure 4: **Visualization of the video denoising steps.** The motion of the person, *e.g.*, his hand, is formed in early stages of the denoising process. the later steps focus on the recovering of the detailed appearance.

the motion generation and mitigate the distribution shift caused by static reference images.

On the other hand, we aim to inject identity while preserving the original prompt following capabilities. Regarding this issue, previous study (Liu et al., 2024) has observed that, the self-attention layers play a crucial role in preserving the geometric and shape details for the identity while the cross-attention layers are primarily responsible for preserving semantic information. Influenced by CustomDiffusion (Kumari et al., 2023), current T2V methods for subject customization choose to finetune the cross-attention layers. However, it tends to undermine the original semantic space, especially when only a single reference image is available, thereby compromising the editability and flexibility of original T2V models. To the end, we propose a Spatial Identity Adapter to inject the identity only into the self-attention layers to preserve the original semantic space. It adopts the residual path of two low-rank matrices including a down block $A^{\text{down}} \in \mathbb{R}^{d \times r}$ and up block $A^{\text{up}} \in \mathbb{R}^{r \times k}$. For the plug-and-play manner, we freeze the pre-trained diffusion model and only insert the adapter into the spatial self-attention layers. Formally, the updated parameter matrices are

$$\tilde{W} = W + \Delta W = W + A^{\text{down}} A^{\text{up}}, \tag{3}$$

for all $W$ in the layers of query, key, and value.

### 4.3 SIMULATED PROMPT AUGMENTATION

To further enhance the robustness of the customization, we introduce simulated prompt augmentation. During the customization process, traditional reconstruction frameworks can only utilize prompts that describe the reference image, which limits the model's generalization capabilities. Benefiting from the non-reconstructive framework, we can incorporate numerous reference-irrelevant prompts during the optimization, *e.g.*, *'V' playing the violin* and *'V' smiling on the beach*. Specifically, we leverage the Large Language Model to create 50 prompts as our simulated prompts with various motion, appearance, and backgrounds and randomly select the prompts during the customization. They align well with actual test scenarios to mitigate overfitting with strong robustness, even when only a single reference image is available.

### 4.4 TRAINING AND INFERENCE

Following previous T2V subject customization methods (Wei et al., 2024), we adopt a two-step training strategy. In the first step, we freeze the video diffusion model and optimize a textual embedding 'V' using Textual Inversion (Gal et al., 2022) to achieve a coarse identity personalization and serve as an initialization. Then we train the spatial identity adapter with ID loss directly on the generated videos to enhance further identity details, which achieves high ID fidelity and preserves original motion dynamics and prompt following. Building on recent advancements in fast sampling methods for T2V models like AnimateLCM (Wang et al., 2024a) , we could generate a high-quality video in only four steps. To preserve the motion dynamic, we inject the spatial identity adapter in the last step during the training. Besides, inspired by previous work [hyperdreambooth], we also use a weight-space loss to mitigate the overfitting and enhance the diversity. Overall, our training loss is as follows:

$$\mathcal{L}_{\text{total}} = \mathcal{L}_{\text{id}} + \lambda \|\hat{\theta} - \theta\|_2^2, \tag{4}$$

where $\lambda$ is the coefficient, $\theta$ and $\hat{\theta}$ are the initial and optimized weights.

During the inference time, we generate videos using the original T2V model with customized textual embedding and our spatial identity adapter. We only introduce our adapter in the final quarter of the denoising steps, which is consistent with the training process.

## 5 EXPERIMENT

### 5.1 EXPERIMENTAL SETTINGS

We utilize the open-source AnimateDiff as our text-to-video generation model. Unless stated otherwise, we use Stable Diffusion 1.5 in conjunction with Realistic Vision (rea, 2023a) during the inference phase. We use ResNet-100 (He et al., 2016) backbone pre-trained on Glint360K (An et al., 2021) dataset as the face encoder. To demonstrate the superiority of our PersonalVideo, we compare it with Magic-Me (Ma et al., 2024), a recent identity-specific T2V customization method, as well as the well-known methods in T2I customization, specifically, LoRA (Hu et al., 2021) with Textual Inversion (Gal et al., 2022) as the initialization.

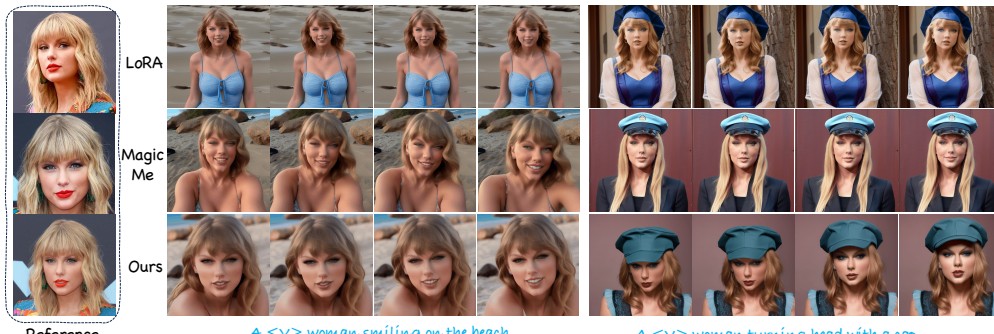

Figure 5: **Qualitative Comparison with Previous Methods.** As observed, both LoRA and Mag-icMe suffer from inferior ID fidelity. Besides, MagicMe has a degradation of prompt following, *e.g.*, *tuning head*. In contrast, our PersonalVideo maintains high ID fidelity and preserve the original motion dynamics and prompt following, which significantly surpasses other methods.

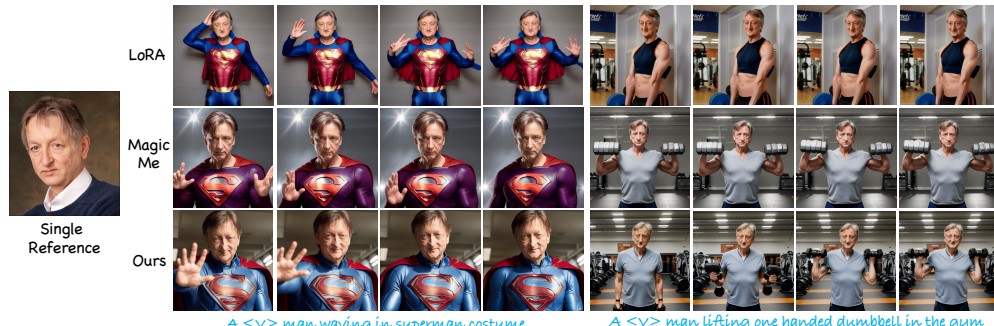

Figure 6: **Qualitative Results for Single Reference.** While LoRA and MagicMe suffer from the inferior ID fidelity and severe degradation of motion dynamics and prompt following, *e.g.lift the dumbell*, our PersonalVideo achieves robust customization with high ID fidelity and preserved motion dynamics and prompt following.

## 5.2 QUALITATIVE RESULTS

We provide a qualitative comparison between PersonalVideo and baselines. As shown in Fig. 5, both LoRA and MagicMe suffer from inferior ID fidelity, due to the tuning-inference gap. Besides, MagicMe has a severe misalignment of the prompt, *e.g.*, *tuning head*. In contrast, our PersonalVideo achieves high ID fidelity and perserves the original motion dynamics and prompt following. To demonstrate the robustness of our method, we also compare for only a single image reference as shown in Fig. 6. The results further underscore the superiority of PersonalVideo, with promising robustness to achieve high ID fidelity and preserve motion dynamics and prompt following.

## 5.3 QUANTITATIVE RESULTS

We present the quantitative results in Tab. 1 and evaluate 1000 generated videos for 20 identities with 50 prompts from these perspectives: (1) Face Similarity: we adopt the ID cosine similar-ity to evaluate ID fidelity, with ID embeddings extracted using Antelopev2 (Deng et al., 2019), which is different from the face recognition models in our framework. (2) Dynamic Degree: we use VBench (Huang et al., 2024), an effective benchmark to compute video dynamics. Besides, we use well-known metrics for video evaluation. As observed, LoRA suffers from inferior face similarity due to the tuning-inference gap. MagicMe gets better face similarity yet degraded dynamics and text alignment. In contrast, our PersonalVideo significantly surpasses previous methods, especially for face similarity and dynamic degree. It achieves high ID fidelity and effectively preserves the ability of the original T2V model, which is consistent with the qualitative results.

## 5.4 COMPATIBILITY WITH CONTROLNET AND STYLE LORAS

| Method | Face Sim. (↑) | Dyna. Deg. (↑) | FVD (↓) | T. Cons. (↑) | CLIP-T (↑) | CLIP-I (↑) |
|---|---|---|---|---|---|---|
| LoRA | 42.62 | 13.86 | 1325.89 | 0.9919 | 26.26 | 44.27 |
| MagicMe | 50.51 | 11.88 | 1336.73 | 0.9928 | 25.48 | 73.03 |
| **PersonalVideo** | **62.35** | **17.80** | **1272.32** | **0.9935** | **26.30** | **76.48** |

Table 1: **Quantitative comparison.** The metrics cover the ability to achieve high ID fidelity (*i.e.*, Face Similarity and CLIP-I), dynamic degree, text alignment (*i.e.*, CLIP-T), distribution distance (*i.e.*, FVD), and temporal consistency.

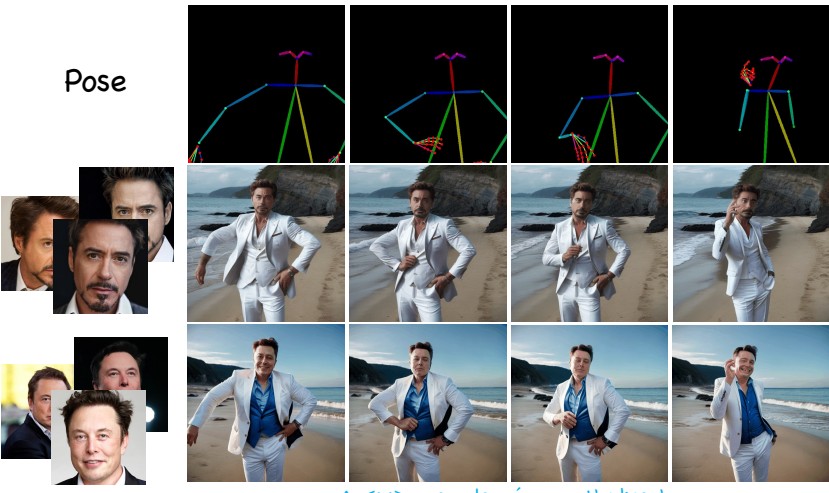

Figure 7: **Controllable generation with ControlNet.** PersonalVideo can be seamlessly integrated with conditional inputs such as poses to generate controllable identity-specific videos.

We showcase that our framework enables controllable generation and demonstrates excellent compatibility with existing fine-grained condition modules, such as ControlNet. As shown in Fig. 7, we effectively utilize ControlNet with reference motion to render identities in the desired poses accurately. It highlights the robust generalization capabilities of our method, which can be seamlessly integrated with existing models.

|  | Face (↑) | CLIP-T | Dynamic (↑) |
|---|---|---|---|
| T2I w Aug | 45.79 | 28.42 | 16.13 |
| T2V w/o Aug | 56.40 | 24.10 | 16.3 |
| T2V w/ Aug | **61.05** | **28.59** | **17.85** |

Table 2: Quantitative ablation study for the **non-reconstructive training** and **simulated prompt augmentation**.

Additionally, we use the Civitai community models to show that it operates effectively with these weights, even though it was not specifically trained on them. The LoRAs we select for this evaluation are separately RCNZ Cartoon 3D (rcn, 2023), GuoFeng RealMix rea (2023b) and GuoFeng (guo, 2023). As shown in Fig. 8, the first row displays the results from the RCNZ Cartoon 3D model, while the second row and the third row highlight the outcomes from the GuoFeng RealMix and GuoFeng model. Our approach consistently delivers reliable facial preservation and effective motion generation, which has great compatibility with these customized style LoRAs.

## 5.5 ABLATION STUDY

**Non-Reconstructive Training.** To validate the impact of proposed non-reconstructive training, we conduct a detailed ablation study in Fig. 9 and Tab. 2. They show the comparison between our PersonalVideo trained on the T2I model or T2V model, including with and without Prompt Augmentation. As observed, tuning on the T2I Model gets inferior ID fidelity due to the tuning-inference gap, which also exacerbates the distribution shift which leads to the blurred background. In contrast, tuning on the T2V model bridges the gap to achieve better ID fidelity with the preserved ability of prompt following. On the other hand, tuning without simulated prompt augmentation overfits the reference images and disrupts the original capability of the prompt following, which manifests as an inability to precisely modify the background with reduced CLIP score. With the introduction of simulated prompt augmentation, this overfitting can be significantly reduced.

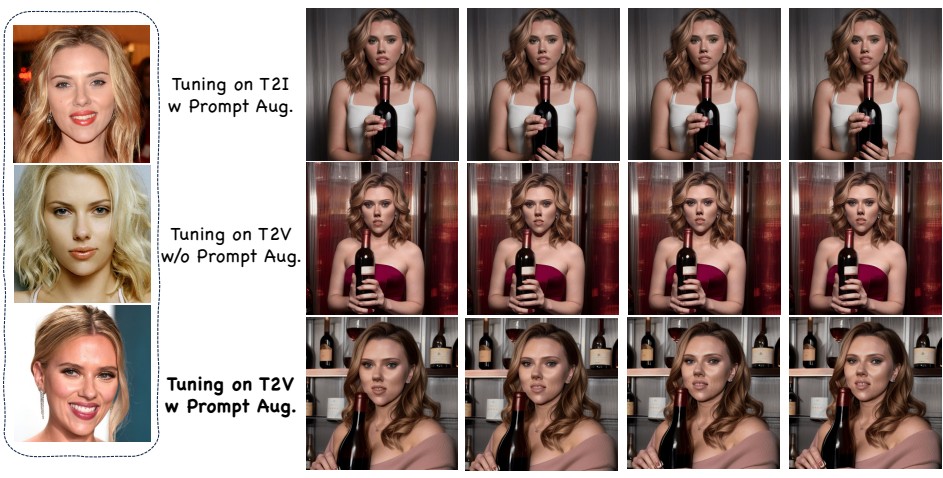

Figure 8: **Compatibility with customized style LoRAs.** We list results from RCNZ Cartoon 3D (rcn, 2023), GuoFengRealMix (guo, 2023) and GuoFeng (rea, 2023b) for the identity.

Figure 9: **Ablation study for the non-reconstructive training and simulated prompt augmentation.** As observed, tuning on the T2I model suffers from inferior ID fidelity and blurred background. Besides, tuning without prompt augmentation degrades the prompt following, *i.e.*, *the wine rack*.

**Different Steps to Inject the Identity.** We also conduct the ablation studies in Fig. 10 and Tab. 3a, which illustrate the improvement in motion dynamics of our Spatial Identity Adapter (SIA) to inject the identity only in the last quarter of denoising steps. As the denoising steps for injecting identity become more concentrated in the later stages, the motion dynamics of the generated videos improve accordingly. This aligns with our observations and validates the effectiveness of our design.

**Different Layers to Inject the Identity.** Besides, Fig. 11 and Tab. 3b demonstrate the improvement in prompt following of our SIA to inject the identity only on the spatial self-attention layer. As observed, injecting only on the cross-attention layer gets inferior ID fidelity with the reference images and disrupts the original capability of prompt following, such as the losing of *exquisite armor*. Although injecting on both self-attention and cross-attention slightly achieves better ID fidelity, it still damages to the prompt following.

| | Face (↑) | Dynamic (↑) | CLIP-T(↑) | | Face (↑) | CLIP-T (↑) | Dynamic (↑) |
|---|---|---|---|---|---|---|---|
| All steps | 62.37 | 13.93 | 26.95 | Cross | 42.68 | 26.20 | 17.70 |
| 1/2 steps | 60.36 | 16.22 | 25.63 | Self + Cross | **62.99** | 23.35 | 17.33 |
| 1/4 steps (Ours) | **63.90** | **18.00** | **27.47** | Self (Ours) | 62.61 | **27.87** | **17.80** |

| (a) Different steps to inject the identity. | (b) Different layers to inject the identity. |
|---|---|

Table 3: Quantitative ablation studies for the proposed **Spatial Identity Adapter**.

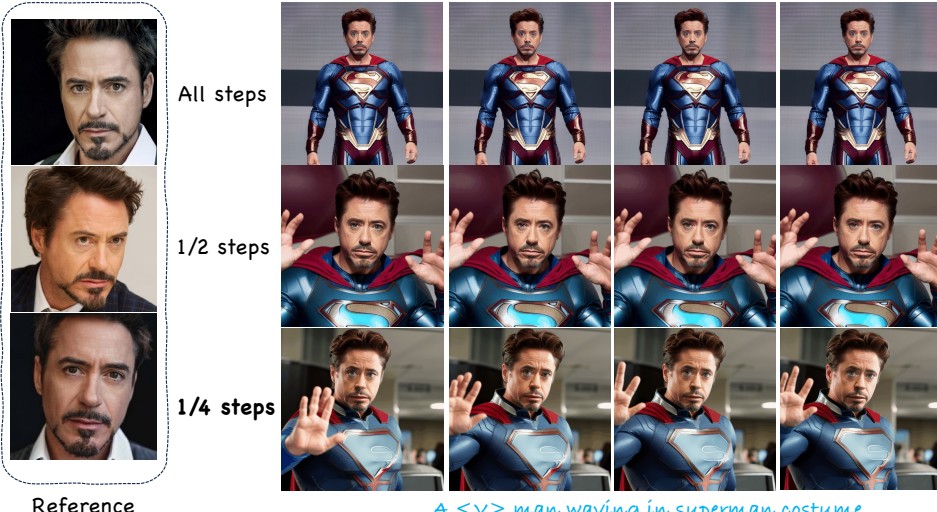

Figure 10: **Ablation for different steps to inject the identity.** As the denoising steps for injecting identity become more later, the motion dynamics of the generated videos improve accordingly.

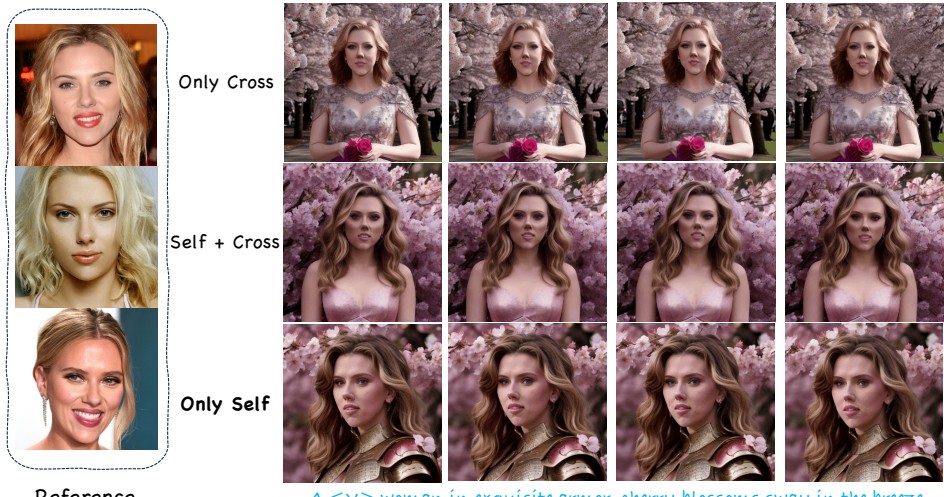

Figure 11: **Ablation for different steps to inject the identity.** As observed, injecting the identity on the cross-attention layer disrupts the ability of prompt following, *e.g.*, the losing of *exquisite armor*.

## 6 CONCLUSION & LIMITATION

In conclusion, we present **PersonalVideo**, a novel framework designed for identity-specific video generation using only a few images, achieving high identity fidelity while preserving the motion dynamics and prompt-following capabilities of the original T2V model. By applying direct supervision on generated videos and introducing a Spatial Identity Adapter, we successfully bridge the tuning-inference gap, mitigating identity degradation. Furthermore, the use of simulated prompts augmentation enhances robustness, allowing for high-quality results even with minimal reference input. Our method demonstrates superior performance over prior approaches, offering a flexible, efficient, and scalable solution for personalized video generation within the AIGC community.

However, our approach still has some limitations. While it enables a plug-and-play injection into the pre-trained T2V model, the results are inherently constrained by the capabilities of the T2V model itself. For example, it fails to generate customized videos that contain multiple identities. One possible solution is to further decouple the attention map of each subject, which will be explored in our future work.

## 7 ETHICS STATEMENT

Our main objective in this work is to empower novice users to generate visual content creatively and flexibly. However, we acknowledge the potential for misuse in creating fake or harmful content with our technology. Therefore, we believe it's essential to develop and implement tools to detect biases and malicious use cases to promote safe and equitable usage.

## 8 REPRODUCIBILITY STATEMENT

We make the following efforts to ensure the reproducibility of PersonalVideo: (1) Our training and inference codes together with the trained model weights will be publicly available. (2) We provide training details in the appendix (Appendix A.1), which is easy to follow. (3) We provide the details of the human evaluation setups in the appendix (Appendix A.2).

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

## A APPENDIX

### A.1 IMPLEMENTATION DETAILS

During training, we optimize the textual token with the learning rate for 1e-3 and the batch size fixed at 4. Each identity training consists of approximately 400 optimization steps. Then we learn the spatial identity adapter for 400 iterations with a learning rate of 1e-4 with the batch size 1. We default to using AdamW optimizer with the default betas set to 0.9 and 0.999. The epsilon is set to the default 1e-8 and the weight decay is set to 1e-2. During inference, we use 25 steps of DDIM sampler and classifier-free guidance with a scale of 7.5 for all baselines. We generate 16-frame videos with $512 \times 512$ spatial resolution and 8 fps. All experiments are conducted on a single NVIDIA A800 GPU.

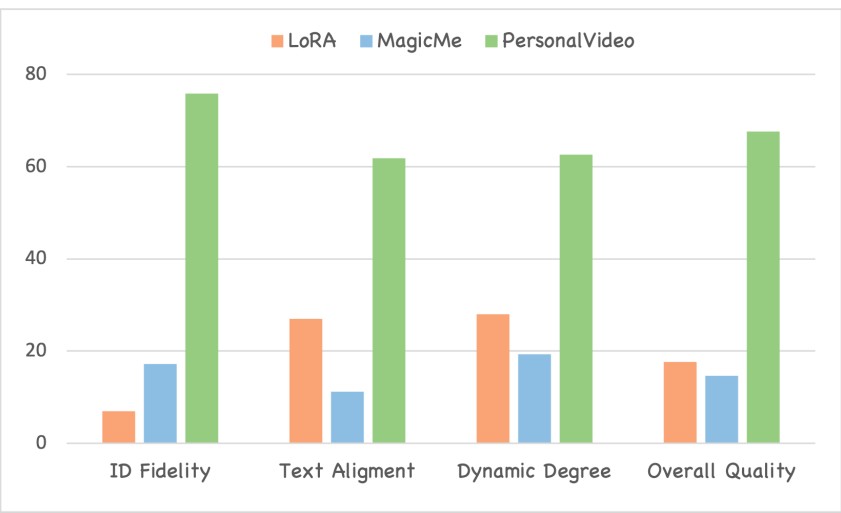

Figure 12: **User Study.** Our PersonalVideo achieves the best human preference compared with other baseline methods.

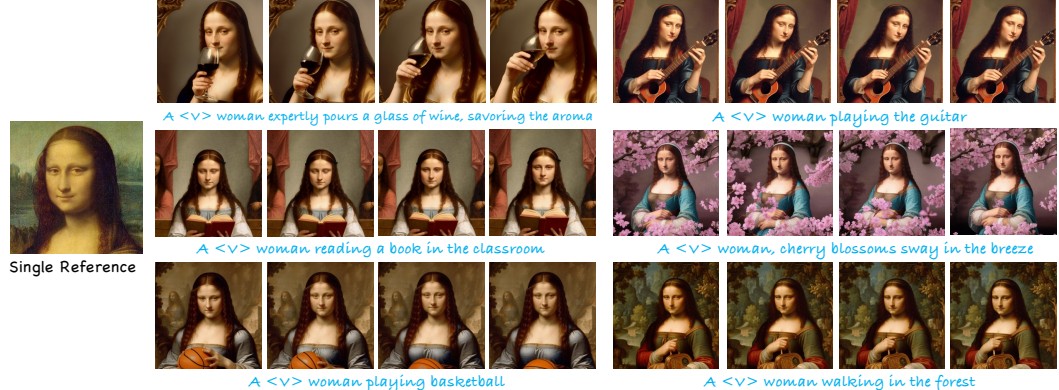

Figure 13: **More results of PersonalVideo** with only just one image.

## A.2 USER STUDY

To further assess the effectiveness of our approach, we perform a human evaluation comparing our method with existing T2V identity customization techniques. We invite 12 people to review 50 sets of generated video results. For each set, we provide reference images alongside videos created using the same seed and text prompt across different methods. We evaluate the quality of the generated videos on four criteria: Identity Fidelity (the resemblance of the generated object to the reference image), Text Alignment (how well the video corresponds to the text prompt), Dynamic Degree (the dynamic degree of motion in the video), and Overall Quality (the overall satisfaction of users with the video quality). As illustrated in Fig. 12, our PersonalVideo receives significantly higher user preference across all evaluation metrics, demonstrating its effectiveness.

## A.3 MORE RESULTS

As shown Fig. 13 and Fig. 14, we present more identity customization results of PersonalVideo, including few or just one reference image. They showcase it achieves high ID fidelity and preserves original motion dynamics and prompt following, which provides further evidence of its promising performance and robustness.

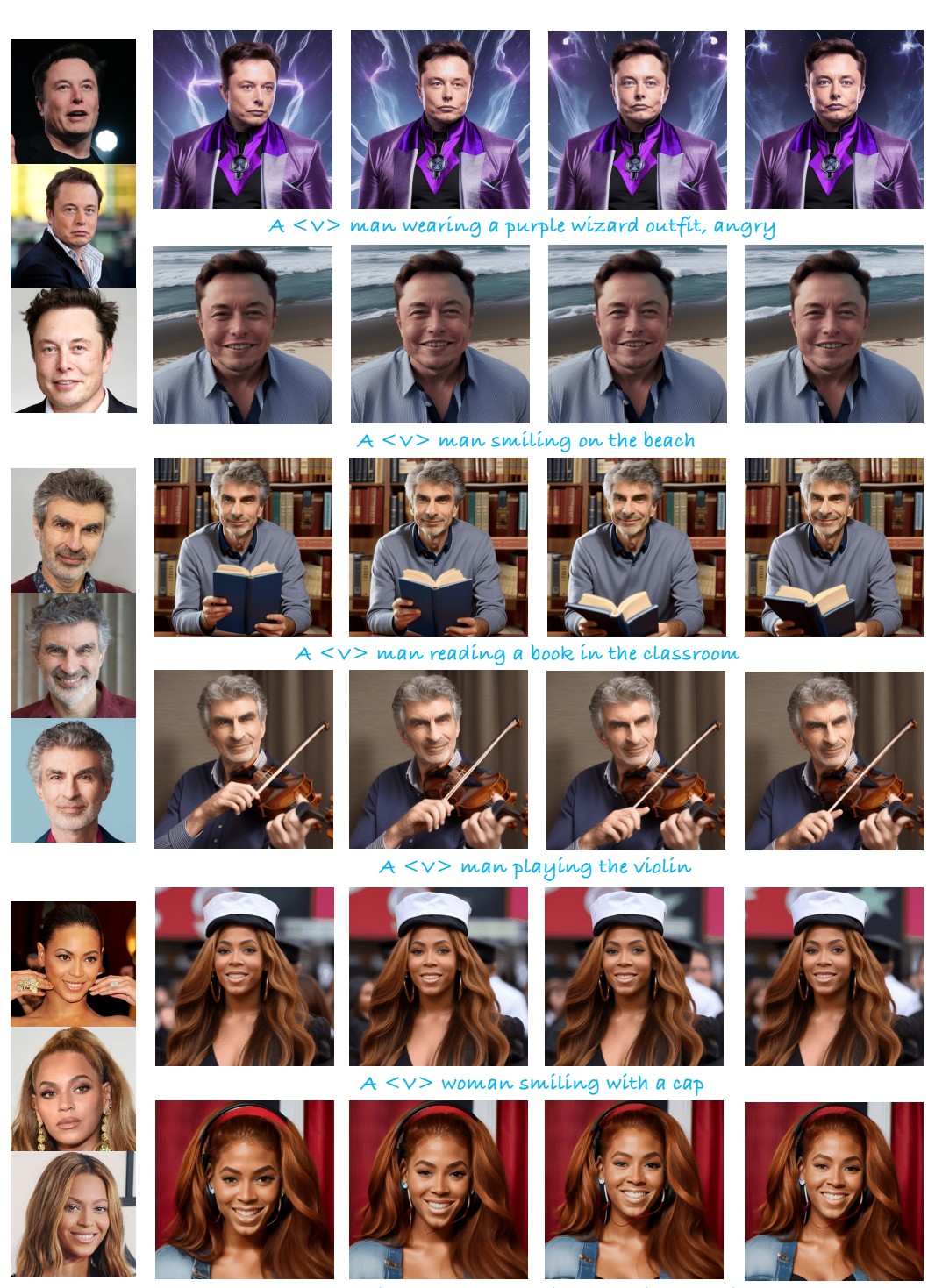

Figure 14: **More results of PersonalVideo** with few images.

