# OpenReview forum: "PersonalVideo: High ID-Fidelity Video Customization With Static Images"
_ICLR.cc/2025/Conference — ICLR 2025 Conference Withdrawn Submission_

### Official Review · Reviewer_Qfa1 · 2024-10-20

**Soundness:** 2
**Presentation:** 1
**Contribution:** 2
**Rating:** 5
**Confidence:** 3

**Summary:**

This paper targets at generating identity-specific videos from static reference images.
To achieve this idea, the authors propose spatial identity adapter, a module to inject the identity information into the spatial self-attention layers during the last denoising steps to achieve both identity fidelity while maintaining motion dynamics from the text-to-video backbone.
Besides, they also introduce direct identity supervision which is applied on generated videos instead of reconstructing images to minimize the tuning-inference gap.
The experiments show that the proposed method achieves better identity fidelity, motion preservation, and visual quality comparing to the previous state-of-the-arts.

**Strengths:**

- The model shows high identity fidelity, while maintaining good visual quality and motion of the original video model.
- The framework can be seamlessly integrated with other components like ControlNet and Style LoRAs to achieve various video generation tasks.

**Weaknesses:**

- Poor writing quality:
    - The article is unclear and ambiguous at many points. Here are a few of examples:
    - Q1: in the formula 2, how do you input the reference image to the model? Do you embed the information of target subject in the text prompt c using textual inversion?
    - Q2: What is the input of spatial identity adapter (SIA)? How can you inject the identity through SIA if it has the exactly same input as self-attention module (as plotted in Figure 3)? Do you input another information, like reference images, to spatial identity adapter?
    - Overall, the paper needs major revisions to be more rigorous.

- Improper claim as a non-reconstructive approach:
    - While the authors claim that their method is a non-reconstructive approach, I disagree this statement in two perspectives:
    - First, the first stage training uses textual inversion to learn a text embedding of the target subject and textual inversion is intrinsically a reconstruction-based learning, which is just like Figure 2(a).
    - Second, the id training objective is a still L2 reconstruction loss between the reference image and a generated video frame.

- Limited contribution:
    - Following the statement above, the proposed method is largely overlapped with the concept of textual inversion. The contribution is simply adding the ID loss and a spatial identity adapter, showing a limited contribution.

- Optimization-based framework:
    - The proposed model is optimization-based framework, which requires training when introducing a new subject. What is the computation to learn a text embedding for a new subject?

- Typos (minor):
    - The title of Figure 11 is incorrect.
    - Line 97~98: "nonconstructive supervision" should be "non-reconstructive supervision"
    - Line 305: failed citation

**Questions:**

- Could you provide the full list of the 50 training prompts (in line 291) and 50 testing prompts (in line 367)? If any training prompts are used during testing, it introduces an unfair evaluation.

- Following the concern above, could you show the results using the prompts which are not the training prompt?

---

### Official Review · Reviewer_boLA · 2024-10-25

**Soundness:** 1
**Presentation:** 2
**Contribution:** 1
**Rating:** 5
**Confidence:** 4

**Summary:**

This paper introduces PersonalVideo, a framework for identity-specific video generation that maintains high fidelity to a given person’s appearance while preserving motion dynamics and prompt alignment. PersonalVideo claims a non-reconstructive approach with direct identity supervision. Key components include the Spatial Identity Adapter, which injects identity at selective denoising stages to enhance consistency and motion, and simulated prompt augmentation Experimental results show that PersonalVideo outperforms models like MagicMe and LoRA in identity fidelity and video quality.

**Strengths:**

1. PersonalVideo introduces a new non-reconstructive framework, using direct identity supervision on generated videos to maintain high identity fidelity and dynamic motion consistency. The task is interesting and the proposed method is simple. The paper presents both qualitative and quantitative results to evaluate the proposed method.

2. PersonalVideo integrates seamlessly with existing T2V components, including ControlNet and style LoRAs, enabling the model to operate across various artistic styles and pose-driven inputs without requiring retraining.

**Weaknesses:**

1. While the paper promotes a non-reconstructive approach, it requires training a text inversion token during inference, similar to previous models like MagicMe. This dependency contradicts the claim of a truly non-reconstructive approach, as it introduces additional training steps that resemble earlier methods.

2. The paper lacks a user study, which is critical for evaluating subjective aspects of video quality, including perceived identity fidelity, motion fluidity, and alignment with prompts. This omission creates a gap in understanding how end users might perceive the effectiveness of the generated videos, potentially overlooking key areas like real-world applicability and user satisfaction.

3. The approach of injecting identity at different steps and layers has been discussed extensively in prior works, such as "P+: Extended Textual Conditioning in Text-to-Image Generation." The lack of clear distinctions from established techniques may make it challenging to demonstrate significant advancements in this area.

**Questions:**

see weaknesses

---

### Official Review · Reviewer_HQUk · 2024-11-02

**Soundness:** 2
**Presentation:** 2
**Contribution:** 2
**Rating:** 5
**Confidence:** 5

**Summary:**

This paper proposes a framework for personalized human video generation. Instead of using reconstruction loss, it applies direct identity supervision on synthesized videos. A learnable Spatial Identity Adapter is further introduced for better identity preservation. Additionally, a prompt augmentation technique is used to reduce overfitting. Experiments have been conducted to evaluate the effectiveness both qualitatively and quantitatively.

**Strengths:**

1. The proposed method can learn personalized video generation from face images.
2. Learning identity through direct supervision on synthesized videos can help avoid the overfitting issue.

**Weaknesses:**

1. The PuLID [1] has already explored adding identity supervision in synthesized images. Given this, the proposed method may not be very novel.
2. Some details in this paper are lacking. For example, the training details and losses for the first step are not provided. Additionally, the details of the baseline (LoRA) training should be included.
3. The difference between the Spatial Identity Adapter and LoRA is unclear. They seem to be the same thing.
4. Some synthesized videos show inferior identity preservation, as seen in Figures 6 and 7.
5. More recent methods, such as [2], should be compared.
6. There are some typos in the paper. For example, the citation of HyperDreamBooth is incorrect in Line 305.

[1] Guo, Zinan, et al. "PuLID: Pure and Lightning ID Customization via Contrastive Alignment."
[2] He, Xuanhua, et al. "ID-Animator: Zero-Shot Identity-Preserving Human Video Generation."

**Questions:**

See weaknesses.

---

### Official Review · Reviewer_BDu7 · 2024-11-03

**Soundness:** 2
**Presentation:** 2
**Contribution:** 2
**Rating:** 3
**Confidence:** 5

**Summary:**

The paper proposes a novel framework, PersonalVideo, for identity-specific video generation using only a few images. The framework achieves high identity fidelity while preserving the motion dynamics and prompt-following capabilities of the original text-to-video (T2V) model. The authors introduce a Spatial Identity Adapter to inject the identity into the T2V model, which bridges the tuning-inference gap and mitigates identity degradation. Additionally, they employ simulated prompt augmentation to enhance robustness and achieve high-quality results even with minimal reference input.

**Strengths:**

1. The proposed framework achieves high identity fidelity, outperforming previous methods in terms of face similarity and dynamic degree.
2. The framework is flexible and scalable, allowing for easy integration with pre-trained T2V models and conditional inputs such as poses.

**Weaknesses:**

1. My main concern about this paper is its contribution and novelty. As mentioned in line 96 of the main paper, "our core insight is applying identity supervision directly to videos generated by the T2V model, thus bridging the tuning-inference gap." However, this core contribution has already been explored by LCM-Lookahead (LCM-Lookahead for Encoder-based Text-to-Image Personalization, Gal et al., ECCV 2024), and the paper does not provide necessary discussions. Therefore, this will greatly weaken the paper's contribution. Regarding the design of the Spatial Identity Adapter, it only references Liu et al. and involves a small contribution of selecting the adapter location. As for the prompt engineering, it is a widely validated method in the community. Overall, the paper's contribution is insufficient.
2. Regarding training and inference, the paper's training process is executed based on fast sampling methods, while inference is performed on the original T2V model. This inevitably reintroduces the tuning-inference gap. The authors need to provide more discussions and explanations, as this is not intuitive.
3. Some related works are missing, such as: DreaMoving: A Human Video Generation Framework based on Diffusion Models, Feng et al., arXiv 2023
4. The fine-tuning stage of the proposed method is complex. The authors need to provide fine-tuning inference time and measure the efficiency of the proposed method.
5. The paper mainly focuses on celebrity images. It is well-known that SD can provide sufficient identity information for them. Have the authors validated the generalizability of the proposed method on non-celebrity images? Furthermore, have the authors considered general video customization, including non-human scenarios? The results are inherently constrained by the capabilities of the T2V model itself, which may limit the quality and diversity of the generated videos.
6. The current framework is limited to generating videos with a single identity, which may not be suitable for scenarios requiring multiple identities.
7. The main text of the paper exceeds 10 pages.

**Questions:**

In conclusion, considering the weaknesses mentioned above, this paper cannot be considered a well-prepared version for ICLR. Therefore, I lean towards rejecting this manuscript.

---

### Note · Authors · 2024-11-15

I have read and agree with the venue's withdrawal policy on behalf of myself and my co-authors.